# Who makes a better university adjustment wingman: Parents or friends?

**Walton Wider** [1]*, **Jem Cloyd M. Tanucan**[2], **Jiaming Lin**[1], **Leilei Jiang**[3], **Lester Naces Udang** [4,5]

**1** Faculty of Business and Communications, INTI International University, Nilai, Negeri Sembilan, Malaysia, **2** College of Teacher Education, Cebu Normal University, Cebu, Philippines, **3** Faculty of Education and Liberal Arts, INTI International University, Nilai, Negeri Sembilan, Malaysia, **4** School of Liberal Arts, Metharath University, Pathumthani, Thailand, **5** Educational Psychology, College of Education, University of the Philippines, Diliman, Philippines

* walton.wider@newinti.edu.my

**Data Availability Statement:** The data underlying the results presented in the study are available from https://doi.org/10.5281/zenodo.8033896.

**Funding:** The author(s) received no specific funding for this work.

## Abstract

The first year of university is one of the most difficult times in a student's life due to numerous changes that occur. This cross-sectional study explores the concept of parental and peer attachment, which has been researched for its ability to predict students' success in higher education. Yet, less research has investigated the mechanisms underpinning the relationship between attachment and university adjustment among first-year students. Hence, the aim of this study was to examine the impact of parent and peer attachment on first-year university students, and understand how these attachments can facilitate university adjustment through identity exploration. This investigation is underpinned by Bowlby and Ainsworth's attachment theory and Arnett's emerging adulthood theory. Data were collected from 568 first-year students at a public university in Sabah, Malaysia, via adapted questionnaires. Structural equation modelling was employed using SmartPLS Software 3.0 to analyse the data. The study found that identity exploration mediates the relationship between parental trust, peer communication, and university adjustment. The findings of this study provide valuable insights for professionals working with emerging adult clients, especially those in higher education institutions, aiming to enhance the adjustment level among first-year students.

## Introduction

Studies on adjustment suggest that it is a diverse and complex phenomenon [1]. According to previous research, students who are able to adjust at university have better academic results, fewer social problems, and less stress [2]. It is the university's priority to provide adequate services and facilities to encourage adjustment among first-year students [3]. The Ministry of Education Malaysia efforts to develop programmes to improve facilities and support programmes are motivated by the fact that many students do not complete their university studies [4]. Despite the ministry's long-standing efforts, the issue of student adjustment, particularly among first-year students, remains a major concern in Malaysia [5]. University life has its own

**Competing interests:** The authors have declared that no competing interests exist.

set of expectations, including but not limited to students' dedication to the institution, their participation in campus-wide events, their general happiness and success as individuals, and their academic performance. Adjustment at a university can be seen by looking at personal qualities or behaviour patterns that have the desired value in an environment [6]. A student is said to be adaptable at a university if he is able to socially integrate with the campus residents, is involved in campus activities, meets academic requirements, and is committed to the higher education institution [7, 8]. If students are unable to adapt to all of the demands of the university environment, they are likely to drop out of their studies. In order to improve adjustment at university, this study focuses on the factors of parent-peer connection as well as the experience of emerging adulthood from the perspective of identity exploration.

The relationship between attachment and identity has been widely studied and supported by numerous researchers. [9, 10] are considered pioneers in the field of attachment theory, which emphasizes the importance of early attachment experiences in shaping an individual's social and emotional development. They proposed that individuals who develop secure attachment styles are more likely to form positive identities and have better social and emotional adjustment throughout their lives. Recent studies have also provided support for the relationship between attachment and identity. [11] conducted a study exploring the relationship between attachment, identity, and academic performance in college students. They found that individuals with secure attachment styles had a stronger sense of identity and performed better academically. Similarly, [12] found that attachment styles were significantly related to identity formation in adolescents. [13] also found that individuals with secure attachment styles had higher levels of identity achievement than those with insecure attachment styles.

Moreover, research has consistently demonstrated the significance of the interplay between attachment and identity in university adjustment. [14] discovered that the combination of attachment and identity predicted college adjustment, specifically in areas of academic achievement and social integration. Similarly, [15] found that attachment and identity interacted to predict university adjustment; individuals with high levels of both constructs experienced the most favorable adjustment outcomes. Other studies have established that attachment is a precursor to identity and, as a result, is linked to heightened levels of adjustment. [16] observed that attachment security was positively associated with identity development during late adolescence, while [17] determined that individuals with secure attachment styles experienced better adjustment outcomes during early adulthood. Additionally, recent studies, such as [18], have reported that secure attachment plays a pivotal role in molding an individual's identity, particularly in terms of self-concept. Finally, [19, 20] conducted studies exploring the relationship between attachment, identity, and mental health outcomes. They found that individuals with insecure attachment styles and poor identity integration had higher levels of anxiety and depression. In conclusion, the research supports the important relationship between attachment and identity, as well as the combination of both constructs on university adjustment. Attachment is a predictor of identity and, consequently, a high level of adjustment, highlighting the importance of early attachment experiences in shaping individuals' development.

Although some studies have looked at the role of identity exploration as a mediator between parental attachment and university adjustment [16, 17, 21–23]. However, the literature review does not suggest investigating identity as a distinct construct with both positive and negative aspects. As a result, this study suggests that the emerging adulthood experience constructs emerge as a form of identity exploration. The main challenge of emerging adults in university, according to [24], is the exploration of identity. There are five emerging adulthood experiences, three of which are positive (identity exploration, self-focus, and a period of possibility) and two of which are negative (instability and feeling "in-between"). As a result, this study

adds to the literature by examining the emerging adulthood experience through the lens of identity exploration as a mediator in the relationship between parental attachment, peer attachment, and university adaptation.

## The importance of parental and peer attachment

Theoretically, university adjustment is frequently linked to Bowlby and Ainsworth's attachment theory, which examines the parent and peer attachment constructs [9, 10]. Many previous empirical studies have investigated the role of parent and peer attachment on university adjustment [22]. According to [25], a separation from parents and identity exploration are also active during this sensitive transition period. This results in an unstable period in which students are exposed to stressful and provocative situations [26]. In comparison to students who have an insecure attachment to their parents, students who have a secure attachment to their parents have a positive view of academics, are motivated, do good academic work, tend to adapt at university, and have the ability to build a protective emotional, behavioural, and social problem [27]. A secure relationship with parents can provide students with a sense of acceptance that allows them to try out a new role and make decisions independently while remaining comfortable knowing that their parents support their behaviour. Previous research has shown that attachment shifts from parents to peers as a person grows older [28]. [9] initially believed that attachment relationships were formed between infants and parents (particularly mothers), but that attachment had shifted to peers as a healthy development among adolescents based on life development. In fact, friendships continue to be important well into adulthood [19]. Peers are more influential in the adjustment process than parents are during the university years [29].

## The importance of identity exploration

Many studies have been discussed in the literature on factors that can influence adjustment at university, and one of them is the exploration of identity [30–33]. [34] proposed that there is a long-term adolescent in a typical industrialised society. [35] observed that many young people today have not yet developed a strong commitment to maturity until the age of 25 or more because they are still actively exploring their identity over a long period of time. This stage is known as emerging adulthood. Most studies agree that entering an institutional structure such as a university increases indicators of well-being among young people [36–40]. The current stage of development at university can foster experiences such as broad opportunities for exploration in a variety of contexts such as cognitive, social, and psychological development, which can influence learning, social activities, and interaction with others [41, 42].

## Adjustment as a life transitional process

For the purpose of this study, [43] model of ecological transition and individual development will be utilised to investigate specifically how first-year college students adapt to their new environments. The research asserted that the transition process can occur in two ways: either separately and in detail, such as the transition from one state/place to another (eg: entry into the university world), or slowly and gradually, such as the developmental or evolutionary transition from one form or stage to other forms or stages (eg: child and adult development). According to this viewpoint, life transitions can be identified in two ways: ecological transitions and individual developmental transitions.

According to [43], individual development occurs slowly and gradually, or from one stage to the next (For example, the emerging adulthood period). During the emerging adulthood period, development takes place primarily in five domains: identity exploration, instability,

self-focus, feeling "in-between," and full of possibilities [24]. Emerging adulthood period is said to be unstable as a result of changes in planning, such as autonomous relationships with parents and romantic relationships [24]. Adjustment is therefore associated with changes in one's ability to understand others and have new ideas when negotiating relationships with parents [44]. These unique emerging adult experiences develop continuously between the ages of 18 and 29 years [24] and serve as the foundation for life development in later adulthood. During this time, students are more actively exploring their identities in order to discover who they truly are. Although identity formation begins in adolescence [34], it is more active and intense during the emerging adulthood period [35]. The period of transitioning also leads to emerging adults becoming more self-focused, i.e., more responsible for themselves and free to make decisions without being influenced by others and not being bound by societal norms in particular [45]. The demands and challenges of emerging adulthood are distinct, and adjustment is associated with achieving a level of self-sufficiency [46]. To achieve the goal of better understanding themselves, emerging adults are given the freedom to explore their potential and live a changing lifestyle.

An ecological transition occurs when a person's situation in the ecological environment changes due to a change in role, place, or both [47]. According to [43], ecological transitions have two characteristics: normative transitions and non-normative transitions. If a person anticipates an ecological transition, that person is said to have a normative ecological transition. According to [35], emerging adulthood normative ecological transitions include work, university, and marriage. These transitions during the emerging adulthood period demonstrate that changes occur not only in terms of personal development but also in terms of location and residence. When an individual does not anticipate a change in conditions or location, a non-normative ecological transition occurs. The ecological transition is normative in the context of this study because adults develop the expectation that adjustment is required during the transition to university. One of the situations that all emerging adults face is the separation from friends due to a change in residence [24]. As a result, emerging adults must understand how to adapt to social networks and feel a part of a new community [48]. Typically, the transition from childhood to adulthood involves not only changes in personal development but also ecological changes that describe an unexpected change in place or role. Individual development and ecological transition, according to [43], must go hand in hand in the transition of life. If the individual does not experience self-development during the ecological transition, the "developmental mismatch hypothesis", which is based on the concept of person-environment, will occur [49]. This concept emphasizes the responsibility of the individual to adapt to a specific or appropriate environment. Rationally, emerging adults in an ecological transition, which in the context of this study is a transition in university, require self-development, which is a period of adulthood that develops concurrently. It is necessary to actively explore one's identity during the transition to university in order to improve one's adjustment at university. If one of the transitions fails, it can result in "developmental mismatch", which can lead to a current crisis at the university. This is due to the individual's development status not matching the roles and demands available in the new and foreign university environment.

## Linking theory of attachment and theory of emerging adulthood

The transition to adulthood is said to present ecological and personal development challenges. Emerging adults experience changes in cognitive, biological, emotional, identity, perspective, relationship, achievement, role, responsibility, and other context [24]. From a developmental perspective, the transition to adulthood is fluctuating. However, [50, 51] see it as a time of excitement and openness. According to [24], the vast majority of young adults are choosing to

continue their education at a college or university. Higher education institutions are the best example of institutions that can offer opportunities and challenges for development that are appropriate for emerging adults [52]. Furthermore, university life is a "social island" distinct from the rest of society, providing a safe haven for emerging adults to explore opportunities in romance, work, and outlook on life [12, 35]. As a result, the transition to the university world can be seen as an ecological transition that requires adults to adapt and take on new roles. Previous attachment theory research mostly focused on the effects of parent and peer attachment on university adjustment. As such, this study connects the theory of attachment and the theory of emerging adulthood in order to better understand and extend the theoretical framework. The researcher created a conceptual framework for the study by combining attachment theory and emerging adulthood theory. This study's conceptual framework as shown in S1 Fig in S1 File suggests a link between parental attachment and peer attachment to individual developmental transitions (emerging adulthood experiences) and then to ecological transitions (university adjustment).

## Method

### Study design

This cross-sectional study targeted first-year students aged between 18 and 25, attending university full-time. A total of 700 survey forms were distributed to first-year students at a public university in Sabah, Malaysia. Of these, 103 were not returned. Upon data entry, 29 surveys were deemed ineligible for use. Sixteen of these were incomplete, and four were excluded because the respondent's age exceeded the predetermined range. Additionally, nine surveys raised concerns of bias in the responses, evident from a pattern that appeared to be straight-lining. Therefore, of the total distributed surveys, 568 were verified as complete, accurate, eligible, and consistent for subsequent analysis. G*Power was utilized to determine the minimum sample size required to achieve statistical power [53]. This study's model includes 4 predictors. Using G*Power with an effect size of 0.15, alpha of 0.05, and power of 0.95, the required minimum sample size was only 129. As a result, our study's sample size of 568 is sufficient, and the results can be confidently reported.

### Study setting

The study was conducted at a public university in Sabah, Malaysia. Data was collected over a two-week period in May 2015.

### Sample and participant recruitment

The snowball sampling method was used to recruit participants. The researcher begins by contacting the university lecturers in identifying a few initial participants who meet the inclusive criteria, such as being in the aged of 18–29 and full-time public university student. Prior to taking the survey, all participants were given an information sheet outlining the purpose of the study, what their participation entailed, and the confidentiality of their responses. Prior to their participation, each participant provided written consent. These initial participants were then asked to invite their friends and classmates who also meet the inclusion criteria by sending them the survey link. The process of recruitment then continues with each participant sharing the link or invitation with their network of contacts who also meet the inclusion criteria, and so on. Data was collected over a two-week period in May 2015. A total of 600 questionnaires were collected, however 32 questionnaires were being omitted due to incomplete responses.

## Instruments

The instruments used in this study are divided into four sections, from Section A to D.

**Section A: Demographic characteristics.** Section A consisted of questions about sociodemographic profiles, such as gender, age, highest level of education, and ethnic group.

**Section B: Student Adaptation to College Questionnaire (SACQ).** The SACQ measurement tool developed by [54] was used to assess university students' adjustment. The SACQ is made up of four sub-scales that assess four aspects of university adjustment (academic, social, personal-emotion, institutional attachment) in first-year students. The SACQ survey tool contains 67 items that use five-point Likert scales (1 = never happens to me, 5 = very often happens to me). Respondents were instructed to respond to the question items by stating how much the statements currently (in the last few days) apply to them. According to the SACQ manual, the institutional adjustment sub-scale contains nine items drawn from the academic (item no. 36) and social (items no. 1, 4, 16, 26, 42, 56, 57, and 65) sub-scales. According to [55], these nine items correlate to the sub-scale of institutional attachment in their research sample of Clark University students. SmartPLS, on the other hand, cannot analyse items that share sub-scales. As a result, the nine items on the institutional attachment sub-scale were not used in this study. While two items with the numbers 53 and 67 are not used in this study, the researcher who created the SACQ questionnaire never explained why these two items were included in the SACQ questionnaire but not linked to the four sub-scales, resulting in a total of 67 items [56].

**Section C: Inventory of Parent and Peers Attachment (IPPA).** IPPA is comprised of three distinct constructs: trust, communication, and alienation. The revised version of IPPA [57], containing 50 items, was used for this study. In the first section, the researcher used 25 items, which asked respondents about their attachment to parents or people who act as parents. Meanwhile, the second section contains 25 items in which respondents are asked how they feel about their best friends. Respondents were asked to rate how true the statement was for them right now. Respondents used five-point Likert scales to express their level of agreement (1—almost never, never true, and 5—almost always, always true). In this particular investigation, we focused solely on the trust and communication dimensions of the IPPA instrument.

**Section D: Inventory of Dimensions of Emerging Adulthood experiences (IDEA).** The IDEA scale is used to assess the identity exploration. [58] created this research instrument, which includes 31 question items. This questionnaire examines respondents' relative identification with five distinct dimensions of emerging adulthood: identity exploration, instability, self-focused, feeling "in-between", and possibility. One dimension, the other-focused was omitted from the analysis. This is due to the fact that this dimension does not belong to the dimension of emerging adult experiences. It was designed to contrast with the "self-focused" subscale. This research tool is scored using four points ranging from 1 (strongly disagree) to 4 (strongly agree). Respondents indicate how much of their current life (as well as the previous few years and the next few years, as they see it) corresponds to each question item.

According to [59], the dimension of identity exploration in the experience of emerging adulthood consists of four dimensions: instability, possibility, self-focus, and feeling "in-between". According to [24], identity exploration is an important part of the emerging adulthood experience because it encompasses all aspects of emerging adulthood experiences. Furthermore, [60] asserted that all dimensions of IDEA are positively and significantly related. As a result of the justification provided, the researcher is able to conceptualise IDEA as identity exploration in this study.

## Pre-test

After the questionnaire has been formulated, the researcher conducts a pre-test to critically evaluate each item before distributing the questionnaire. It is essential to conduct pre-testing to identify potential issues respondents may have understanding or defining questionnaire questions [61]. A small sample of respondents is used for pre-testing to ensure that the questionnaire questions are clear and easily understood. This is done to reduce problems such as ambiguous and biased wording, as well as to test the questionnaire's appropriateness in the context study [62]. Convenience sampling was used to select five first-year students to participate in this pre-test. The participants of this pre-test were excluded from the study sample. In addition, two academicians who are experts in the field of developmental psychology have been chosen to provide suggestions and advice for improving the quality of measurement instruments. Each respondent receives a copy of the questionnaire and is instructed to respond, identify any difficulties in understanding the questions and instructions presented on the questionnaire, and evaluate the sentence's relevance to the intended meaning. Following this, modifications were made: 1) Responses in Section A were refined to specify the specific field of study rather than broad fields of study; 2) Question numbering was restructured based on feedback regarding the item list's length; and 3) A culturally-sensitive question in Section B was realigned. The researcher identifies and reexamines every comment and suggestion made.

## Data analysis

This study applies PLS-SEM to gain greater insights into the mediating effect of identity exploration on the relationship between parental trust, parental communication, peer trust, peer communication, and university adjustment. We used PLS-SEM because of its inherent suitability for exploratory studies, which could examine both the measurement and structural models [63], using the SmartPLS 3.0 [64] software package. SmartPLS is well-suited for analyzing data from small or moderate sample sizes, which can be common in social science research [65, 66]. It uses a partial least squares (PLS) algorithm, which is more robust than traditional maximum likelihood estimation methods for smaller sample sizes [67]. SmartPLS is capable of analyzing complex models with multiple latent variables and observed variables [68]. It can also handle non-linear relationships and interactions between variables, which can be difficult to analyze using traditional methods [65]. Additionally, SmartPLS is able to estimate and test complex models that include multiple mediators and non-linear relationships between variables. For example, [64] demonstrated the use of SmartPLS to estimate a mediation model with multiple mediators and non-linear relationships between variables. For the mediator analysis, the researcher employs the product of the coefficient's method with bootstrapping [69]. In addition, demographic factors collected in the study were also tested as possible control variables.

## Results

Table 1 depicts the demographics of the 568 participants in this study. The majority of respondents are 21 years old (54.9 percent), followed by 20 years old (28.9 percent), 22 years old (6.7%), 19 to 23 years old (4.0%), 24 years old (0.7%), 25 years old (0.5%), and 18 years old (0.2%). The majority of respondents (66.2%) possess a Malaysian Higher School Certificate (Malay: *Sijil Tinggi Persekolahan Malaysia*), followed by the matriculation level (27.1%), the diploma level (6.0%), and the foundation level (0.7%). In terms of gender, women account for 63.7% of respondents, while men account for 36.3%. The majority of respondents are Bumiputera Sabah (48.8%), followed by Malays (23.1%), Chinese (18.1%), others (4.2%), Bumiputera Sarawak (4.0%), and Indians (1.8%). In terms of the participants' Grade Point Averages

**Table 1. Demographic profile of respondents (N = 568).**

| Variables | Frequency | % |
|---|---|---|
| Age | | |
| 18 | 1 | 0.2 |
| 19 | 23 | 4.0 |
| 20 | 164 | 28.9 |
| 21 | 312 | 54.9 |
| 22 | 38 | 6.7 |
| 23 | 23 | 4.0 |
| 24 | 4 | 0.7 |
| 25 | 3 | 0.5 |
| Highest level of education | | |
| STPM | 376 | 66.2 |
| Diploma | 34 | 6.0 |
| Matriculation | 154 | 27.1 |
| Foundation | 4 | 0.7 |
| Gender | | |
| Male | 206 | 36.3 |
| Female | 362 | 63.7 |
| Ethnic | | |
| Malay | 131 | 23.1 |
| Chinese | 103 | 18.1 |
| Indian | 10 | 1.8 |
| Bumiputera Sabah | 277 | 48.8 |
| Bumiputera Sarawak | 23 | 4.0 |
| Others | 24 | 4.2 |
| Grade Point Average (GPA) | | |
| 3.50–4.00 | 62 | 10.9 |
| 3.00–3.49 | 258 | 45.4 |
| 2.50–2.99 | 213 | 37.5 |
| 2.4 and below | 35 | 6.1 |
| **Field of Study** | | |
| Humanities | 120 | 21.1 |
| Psychology | 197 | 34.7 |
| Science | 143 | 25.2 |
| Business | 54 | 9.5 |
| Engineering | 34 | 6.0 |
| Computing and Informatics | 12 | 2.1 |
| Agriculture | 2 | 0.4 |
| Food Science | 6 | 1.1 |

(GPA), 10.9% had a GPA ranging between 3.50 and 4.00 (n = 62). The majority of the participants, 45.4%, had a GPA in the range of 3.00–3.49 (n = 258). This was followed by 37.5% of the participants who had a GPA between 2.50 and 2.99 (n = 213). Lastly, a smaller proportion, 6.1%, had a GPA of 2.4 and below (n = 35). The study's participants came from a variety of fields of study. Psychology had the largest representation, with 197 participants accounting for 34.7% of the sample. Science had 143 participants, accounting for 25.2% of the sample. There were 120 participants (21.1%) in the Humanities field. The business category had a total of 54 participants, accounting for 9.5% of the sample. Less popular fields included Engineering

(n = 34, 6.0%), Computing and Informatics (n = 12, 2.1%), Food Science (n = 6, 1.1%), and Agriculture (n = 2, 0.4%).

## Measurement model assessment

PLS-SEM encompasses both the outer model, referred to as the measurement model, and the inner model, known as the structural model [70]. In assessing the measurement model, we analyzed indicator reliability and validity to ensure the measures used were appropriate [71]. This evaluation included verifying item reliability, ascertaining construct's internal consistency, and establishing both convergent and discriminant validity [72].

Six reflective constructs were used in this study's framework: parental trust, parental communication, peer trust, peer communication, identity exploration, and university adjustment. A number of measures, including composite reliability (CR) and rho A, were used to assess the measurement model's construct reliability. These criteria must be greater than 0.7. [72]. Furthermore, to establish convergent validity, the extracted average variance (AVE) should be greater than 0.5 [73]. Previous research has frequently argued and asked questions about "how many items are appropriate to use to represent a construct in SEM analysis?" According to [74], using a few indicators (items) in a study is sufficient, whereas using more than three indicators on a latent variable is less desirable because adding too many indicators provides little benefit in a study, whereas using two items in SEM is reasonable. A small number of items is thought to be the best option because it allows for the development of a parsimonious model with simple explanatory power, increasing the likelihood of model fitness. As a result, it is suggested that the most appropriate items, those with a high loading, be retained. As a result, 58 items from the university adjustment construct, 18 items from identity exploration, 3 items from parental trust, 5 items from parental communication, 2 items from peer trust, and 3 items from peer communication had low loadings and were deleted. As shown in Table 2, the results of the measurement model assessment using the criteria met the required thresholds, indicating that the study model has reliability and convergent validity.

The researcher used the heterotrait-monotrait (HTMT) ratio to determine the discriminant validity of the constructs in this study [73]. When the HTMT ratio is less than 0.85 or 0.90, discriminant validity is achieved [75]. Table 3 shows that the HTMT value for all constructs is less than 0.90, indicating that discriminant validity for this study model has been achieved.

## Structural model assessment

For the structural model, we conducted the direct and indirect effects analysis to test the hypothesis relationship. This assessment requires examining the R-squared ($R^2$) value, significance of indirect effects through the t-value, a 95% confidence interval (CI 0.95), and the effect size ($f^2$) [69, 72]. In addition, GPA was incorporated as a control variable within the structural model to account for its potential confounding effects on university adjustment.

Firstly, to make sure that there were no issues with lateral collinearity in the structural model, we conducted an analysis of the collinearity between the study variables. Table 4 shows that the VIF values of all constructs were less than 5 (ranged from 1.000 to 2.564), indicating that there was no multi-collinearity problem. $R^2$ values for identify exploration and university adjustment were 0.180 and 0.188, respectively, which are acceptable for behavioural science studies [75]. In the analysis of direct effects as depicted in S2 Fig in S1 File, the results shows that parental trust had a significant positive effect on identity exploration ($\beta$ = 0.157, t = 2.162). Similarly, peer communication was found to have a positive effect on identity exploration ($\beta$ = 0.157, t = 2.162). Moreover, identity exploration significantly influenced university adjustment ($\beta$ = 0.433, t = 11.542). Contrastingly, parental communication's effect on identity

**Table 2. Measurement model assessment results.**

| Construct | Loadings | rho_A | CR | AVE |
|---|---|---|---|---|
| University Adjustment | | 0.861 | 0.886 | 0.528 |
| ACAD19 | 0.631 | | | |
| ACAD2 | 0.778 | | | |
| ACAD7 | 0.736 | | | |
| INS1 | 0.772 | | | |
| SOC1 | 0.676 | | | |
| SOC4 | 0.694 | | | |
| SOC6 | 0.783 | | | |
| Identity Exploration | | 0.897 | 0.913 | 0.514 |
| SF5 | 0.658 | | | |
| FIB2 | 0.685 | | | |
| FIB1 | 0.686 | | | |
| SF2 | 0.689 | | | |
| IE5 | 0.700 | | | |
| POSS4 | 0.703 | | | |
| IE7 | 0.717 | | | |
| SF4 | 0.762 | | | |
| IE4 | 0.770 | | | |
| IE6 | 0.788 | | | |
| Parental Trust | | 0.898 | 0.910 | 0.591 |
| PA_TRUST1 | 0.828 | | | |
| PA_TRUST10 | 0.787 | | | |
| PA_TRUST2 | 0.827 | | | |
| PA_TRUST4 | 0.77 | | | |
| PA_TRUST6 | 0.704 | | | |
| PA_TRUST7 | 0.765 | | | |
| PA_TRUST9 | 0.69 | | | |
| Parental Communication | | 0.822 | 0.867 | 0.621 |
| PA_COMM1 | 0.841 | | | |
| PA_COMM3 | 0.824 | | | |
| PA_COMM6 | 0.698 | | | |
| PA_COMM8 | 0.783 | | | |
| Peer Trust | | 0.915 | 0.929 | 0.619 |
| PE_TRUST10 | 0.820 | | | |
| PE_TRUST2 | 0.813 | | | |
| PE_TRUST3 | 0.823 | | | |
| PE_TRUST4 | 0.775 | | | |
| PE_TRUST5 | 0.802 | | | |
| PE_TRUST6 | 0.777 | | | |
| PE_TRUST7 | 0.744 | | | |
| PE_TRUST9 | 0.738 | | | |
| Peer Communication | | 0.855 | 0.885 | 0.607 |
| PE_COMM1 | 0.813 | | | |
| PE_COMM2 | 0.808 | | | |
| PE_COMM3 | 0.816 | | | |
| PE_COMM7 | 0.725 | | | |
| PE_COMM8 | 0.728 | | | |

**Table 3. Discriminant validity through $HTMT_{0.90}$.**

| | | 1 | 2 | 3 | 4 | 5 | 6 |
|---|---|---|---|---|---|---|---|
| 1 | Identity Exploration | | | | | | |
| 2 | Parental Communication | 0.321 | | | | | |
| 3 | Parental Trust | 0.348 | 0.897 | | | | |
| 4 | Peer Communication | 0.393 | 0.403 | 0.382 | | | |
| 5 | Peer Trust | 0.395 | 0.383 | 0.451 | 0.885 | | |
| 6 | University Adjustment | 0.488 | 0.423 | 0.472 | 0.429 | 0.408 | |

exploration was not statistically significant ($\beta$ = 0.055, t = 0.835). The same was true for the relationship between peer trust and identity exploration ($\beta$ = 0.153, t = 1.913). In examining the mediation effects, our results highlighted the significance of indirect pathways. Specifically, parental trust had a significant indirect effect on university adjustment through identity exploration ($\beta$ = 0.073, t = 1.937). Similarly, the mediating role of identity exploration in the relationship between peer communication and university adjustment was also significant ($\beta$ = 0.068, t = 2.048). This underscores the pivotal role identity exploration plays in mediating the relationship between both parental trust and peer communication, and university students' adjustment outcomes.

## Control variable

In addition to the testing of the proposed links between exogenous and endogenous latent variables as shown in the structural model, one control variable was also examined in this study. GPA added as a control variable in the final model. Control variable is treated as exogenous latent variable similar to other exogenous variables in the model [76]. Control variable should be included for the expressed purpose of accounting for known or potential confounding effects on any construct in the model [77]. GPA is often used as a measure of academic ability or preparedness. Students with higher GPAs are generally perceived to have better academic skills and knowledge, which may make them better equipped to handle the academic demands of university. Therefore, controlling for GPA in studies investigating university adjustment helps to ensure that any observed effects are not simply due to differences in academic ability or preparation. Secondly, GPA has been found to be associated with various aspects of university adjustment. For example, students with higher GPAs tend to experience less academic stress, have better study habits and time management skills, and are more likely to persist and succeed in their studies. Therefore, controlling for GPA helps to isolate the effects of other

**Table 4. Results of hypothesis testing.**

| Direct/ Indirect Effect | Path Coefficient | t-value | 95% Bias-Corrected Confidence Interval | VIF | Supported |
|---|---|---|---|---|---|
| Parental Trust → Identity Exploration | 0.168 | 2.085 | [0.009, 0.323] | 2.429 | Yes |
| Parental Communication → Identity Exploration | 0.055 | 0.835 | [-0.073, 0.180] | 2.311 | No |
| Peer Trust → Identity Exploration | 0.153 | 1.913 | [-0.016, 0.295] | 2.564 | No |
| Peer Communication → Identity Exploration | 0.157 | 2.162 | [0.022, 0.305] | 2.564 | Yes |
| Identity Exploration → University Adjustment | 0.433 | 11.542 | [0.351, 0.501] | 1.000 | Yes |
| Parental Trust → Identity Exploration → University Adjustment | 0.073 | 1.937 | [0.002, 0.150] | | Yes |
| Parental Communication → Identity Exploration → University Adjustment | 0.024 | 0.831 | [-0.034, 0.078] | | No |
| Peer Trust → Identity Exploration → University Adjustment | 0.066 | 1.871 | [-0.007, 0.131] | | No |
| Peer Communication → Identity Exploration → University Adjustment | 0.068 | 2.048 | [0.009, 0.139] | | Yes |

variables on university adjustment that are not simply due to differences in academic performance. In order to test for the effects of control variable in this study GPA was included in the model and linked to university adjustment. The bootstrapping was applied to see the relationship between the control variable and university adjustment. The bootstrapping result indicates GPA has insignificant relationship with university adjustment ($\beta = 0.004$, t = 0.126), suggesting that identity exploration has no change as the education level change.

## Discussion

The transition to university brings with it a variety of life demands that can interfere with students' adjustment to university life, particularly among first-year students. As discussed, previous research indicates that the attachment to parents and peers is an important factor that can influence adjustment during the transition to the university. This study introduces the identity exploration as a mediator in an effort to examine how first-year students utilize attachment with parents and peers to achieve identity exploration and further improve university adjustment. The study's findings show that identity exploration acts as a mediator in the relationship between parental trust and university adjustment, but it does not act as a mediator in the relationship between parental communication and university adjustment. In contrast, the findings indicate that identity exploration mediates the relationship between peer communication and university adjustment. However, identity exploration did not act as a mediator in the relationship between peer trust and university adjustment. This study has empirically linked attachment theory and emerging adulthood theory based on the assertions of [43]. According to the findings of this study, parent and peer attachment have similar goals but serve different functions in enhancing identity exploration. Aspects of trust were found to be important in attachment relationships with parents, while communication aspects were found to be important in attachment relationships with peers.

This study provides strong support for the argument that starting university is akin to a 'strange situation' [25, 78]. Emerging adults require a primary attachment subject (parents), just like infants, to explore new environments. This study also supports [79] argument that university students who have a secure attachment relationship with their parents are better able to adjust to developmental tasks at university. The question also arises as to why the identity exploration does not act as a mediator in the relationship between the communication and adjustment subscale in university. According to [57], the behaviour of separation and reunion is evaluated to measure the level of attachment in babies; for adults, the assessment of the level of attachment is seen in terms of the cognitive-affective dimension. This study operationalised the communication sub-scale as "the level and quality of verbal communication with parents". During the transition to college, direct communication with parents diminishes, and parents no longer constantly monitor their children. The decrease in verbal communication between students and parents makes parents unable to respond to or be sensitive to students' emotional states, interfering with the process of student identity development. However, it should be emphasized that even if students have autonomy from their parents, this does not imply that the level of attachment is weak. According to [80], students who live away from their families during the transition period to the university world have a better attachment relationship with their parents than students who live with their parents. The results of this research support this notion, showing a strong association between trust in parents and not for communication with them. According to [57], emerging adults view trust as "perceived security" provided by the main attachment subject (parents). In this study, trust is operationally defined as "respect for parents and mutual trust". Students maintain strong bonds by demonstrating mutual respect and trust. Mutual trust between students and parents leads to the exploration of an

active and healthy environment, which leads to a good adjustment during the transition to first year at university [81, 82]. This study's findings support the claims that transition and adjustment at university are important developmental milestones for students because they begin their journey into emerging adulthood at this time [35, 83–85]. The transition to university life places a high demand on first-year students for autonomy and responsibility. A successful transition to the university environment necessitates students' ability to manage time effectively, create a manageable academic and social schedule, complete academic tasks, consult about a new social life, and respond to new challenges and pressures [86, 87]. This period also includes an extension of the search for self-definition and identity for students in particular [88]. This study also demonstrated that a close relationship between emerging adults and their parents while at university can aid in student development by providing a "safe base" that allows students to explore and build capacity in a life environment outside of the family environment, which in turn can have a positive influence on adjustment to the university in particular.

Studies of emerging adults indicate that during the transition to university, the majority of students will leave home and separate from their parents; this prompts students to seek support from peers to assist them in adjusting to this significant life change [89, 90]. Attachment theory emphasizes that infants have attachment relationships with one primary subject, who is typically the mother, while children begin to develop attachment relationships with other family members such as siblings or caregivers, and at the adolescent stage and beyond, individuals begin to develop attachment relationships. a diverse group made up of people who are not family members, such as peers. Indeed, attachment theory emphasizes the importance of the quality of the attachment relationship, which is not limited to mothers, fathers, and grandparents. In fact, attachment to others is viewed as not causing any major issues or problems in emerging adults. In contrast to peers, the attachment function is more of a "safe haven," in that it helps adults find comfort in the face of threats or fear. Although parent and peer attachment serve different functions, they both serve the same purpose: to be a factor that can assist first-year students in dealing with the transition of identity development and in adjusting to the new university environment. Peers serve as a safe haven, acting as individuals who can provide relief and assistance through the challenges of development and adjustment at university. According to [91], time spent with parents begins to decline in emerging adulthood period. Because parents are currently unable to interact face-to-face with students, peers are the closest individuals who can be used as a safe space to express concerns and opinions. According to [92], during the growing adulthood period, students tend to turn to peers in desperate situations. Aspects of good communication quality, in particular, can influence the exploration of student identity, where students can share all of their problems and concerns. This study backs up [93] claim that good communication with friends indicates the presence of secure relationships with peers, thereby supporting the importance of communication in attachment. Communication with close friends during the period of developing adulthood plays a crucial role in helping adults develop toward identity formation and, at the same time, influences adaptation in the university [94, 95].

The study's findings indicate that the aspect of trust in parents has a significant indirect relationship with university adjustment. People dealing with university students such as lecturers, academic advisers, counsellors, and university administrators must understand that parental trust and peer communication must be emphasised in order to encourage students to actively explore the university environment. Starting university requires students to become independent, so they rarely see their families. For this reason, it is crucial that students and parents maintain a level of trust in one another so that college students can confidently explore unfamiliar environments without feeling threatened or uncertain. Additionally, first-year

students spend more time with their peers as a result of their independence from their families. Thus, good communication elements must always exist between students and their peers so that students can explore new environments and then make adjustments at university. The university should have a programme that can implement bond formation with parents and peers, particularly in the area of parental trust and peer communications, so that students are more aware of the need to maintain and preserve existing good bonding relationships despite having achieved autonomy from parents. Peers, for example, must function as individuals who can provide a safe space for students to express their problems and concerns while at university. When students discuss problems, their peers must be sensitive and responsive to the opinions expressed. Peers are also encouraged to be more sensitive to things that may upset students' emotions and to be more open in communicating about university problems. Furthermore, the study's findings suggest that students should be more active in discovering their identity, developing self-awareness, determining their own values and beliefs, and learning to plan for the future. Additionally, students are reminded to have an open mind and to not be afraid to try new things and experiment with different things. Students are also advised to mature gradually, to think optimistically, to be independent, to focus on themselves, to be responsible, and to discover their own abilities. Students should also be aware that the emerging adulthood experience at university can be unstable, with students experiencing confusion, limited feelings, high stress, and anxiety. This is a common side effect of active identity exploration. If the student actively engages in the suggested exploration, he or she will acquire soft skills and become a holistic graduate, that is, a person with good knowledge, morals, behaviour, mindset, and manners as a result of the university learning experience [96].

While the current study was conducted prior to the advent of COVID-19 in 2015, its emphasis on attachment and identity for university adjustment aligns with more recent findings in the context of the pandemic. During the COVID-19 pandemic, [97] investigated changes in parent and peer attachment and the associated adjustment challenges among college students in the United States. They discovered that students reported lower attachment security, which was associated with greater adjustment difficulties. Reduced peer attachment, in particular, was associated with feelings of loneliness and depression. Parental attachment, on the other hand, appeared to balance the pandemic strains. A notable finding from their study was a stronger link between lower parental attachment and feelings of loneliness among Asian American students, possibly indicating cultural differences in family interdependence.

[98] investigated the relationship between attachment to parents and peers, as well as the psychological consequences of the pandemic in late adolescents. Their findings highlighted the significant impact of attachment to fathers and peers, rather than mothers, on adolescent distress during COVID-19. The study also highlighted the link between high levels of alexithymia (difficulty in recognizing and expressing emotions) and emotional-behavioral difficulties. Connecting these studies to ours reveals a recurring theme: while attachment patterns influence psychological well-being and adjustment, factors such as identity (as investigated in our study) and alexithymia (as investigated in Tambelli et al.'s research) can mediate these relationships. These mediating variables may provide additional insight into the complexities of how young adults navigate challenges, such as transitioning to university or dealing with unprecedented global events such as a pandemic. In summary, the significance of attachment and its relationships with other psychological constructs like identity continue to be vital in understanding university adjustment and overall psychological well-being, regardless of the diverse socio-historical contexts and research goals.

The implications of our study are twofold, both theoretical and practical. Theoretically, this study provides robust empirical evidence that experiences during emerging adulthood, particularly identity exploration, can mediate the relationships between parental attachment, peer

attachment, and university adjustment. Building on the perspectives of [99], our findings furnish a comprehensive conceptual framework. Further, this research establishes an empirical connection between Bowlby's attachment theory and Arnett's theory of emerging adulthood. It elucidates how these two theories intersect and influence each other. While attachment theory underscores the role of attachment in identity exploration, our results indicate that parent and peer attachment bolster identity exploration via distinct mechanisms, such as trust and communication. Moreover, this research amplifies the framework of identity by presenting it as a multifaceted construct, echoing the sentiments of [59]. On the whole, this research enhances existing literature by scrutinizing the mediating role of identity exploration between parental and peer attachment and university adjustment.

The practical implications derived from our study are profound. Firstly, the importance of parental trust and peer communication in facilitating the transition for first-year college students cannot be overstated. This suggests that educational policymakers, especially within Malaysian universities, ought to prioritize the development and implementation of comprehensive transition programs that accentuate these elements. Such strategic policies could pave the way for platforms or initiatives that actively nurture the formation of supportive peer relationships. Recognizing the vital role of strong friendships and open peer communication, these initiatives would be instrumental in creating an academic atmosphere where students feel sufficiently secure to voice their concerns and challenges, thereby enhancing their overall well-being and social adjustment. Furthermore, the significant influence of parental trust on university adjustment beckons the formulation of policies that bolster parental engagement. Initiatives might encompass workshops, seminars, or even counselling sessions tailored to amplify the trust between students and their parents. With fortified trust, students are more likely to confidently embrace new roles, make independent decisions, and navigate the myriad challenges university life presents. Lastly, understanding the pivotal mediating role of identity exploration, it becomes imperative for policymakers to advocate for curricular and extracurricular platforms geared towards this end. Such avenues, offering experiences from internships to cultural immersion programs, would serve as fertile grounds for students to introspectively explore their personal, political, religious, and professional identities. In synthesizing these policy implications, it becomes evident that university governing bodies, in collaboration with educators and counselors, can enact impactful measures that promote a holistic student development, equipping them with the tools to address both academic and personal challenges head-on.

## Limitations and future research

Despite the fact that the design of this study was chosen based on its suitability to address the study's objectives and key elements, there are still some limitations that the researcher cannot avoid. The first is related to the concept of parental attachment; in this study, respondents are asked to answer questions about the influence of parental attachment without making any distinction between mothers and fathers. According to [100], the family environment as a whole is more important than the specific relationship with the mother or father in determining the emerging adult's sense of social competence. Among the studies that use a single assessment of parents is the study of [101], who justifies this by stating that it is possible unless the respondent has a "very different" relationship with the mother and father, which requires them to assess the mother and father differently. Thus, future research can expand on this study to examine mother and father attachments separately.

Second, the study's limitation related to generalizability is justified by the sampling method and the specific population that was studied. While snowball sampling can be an effective way

to recruit participants with specific characteristics, it can also introduce biases into the sample, leading to potential limitations in generalizing the results to other populations. Furthermore, the study was limited to first-year students at one Malaysian institution of higher education. This means that the sample may not be representative of other populations, such as students at different institutions or at different stages of their academic career. Additionally, the cultural context and educational system in Malaysia may differ from other countries, further limiting the generalizability of the results. Therefore, while the findings of the study may be informative and relevant for the specific population and context that was studied, caution should be taken when attempting to generalize the results to other populations or contexts. Nevertheless, the findings of this study as a whole provide new and profound insight and understanding of parental and peer attachment, identity exploration, and university adjustment in the context of first-year students. The researcher believes that assessing this research model from a global perspective by comparing countries and cultural contexts would make this study more interesting. Furthermore, the sample can be drawn by distinguishing the year of study and not limiting the study to first-year students. Previous research has demonstrated that adjustment at university can differ depending on the year of study.

## Conclusions

The results of this study demonstrate that the attachment theory and the emerging adulthood theory each have their own unique connections during the emerging adulthood period. This study investigates the relationship between parental and peer attachment in the context of the transition to first-year university, as well as the benefits of identity exploration for first-year students. The study discovered that identity exploration serves as a mediator between parental trust and university adjustment, as well as between peer communication and university adjustment. In conclusion, those who work with university students in particular should be able to identify how factors like parental attachment can influence a successful transition to university life. People who work with emerging adults must also recognize the context of identity exploration, which can influence the relationship between attachment and adjustment during the transition to university life. Furthermore, this study found that relationships with peers are just as important as those with parents, though they can impact adjustment in different ways. Overall, the results of this study lend credence to the notion that first-year students' attachment to both family and friends plays a crucial role in the success of their transition to university life.

## Supporting information

**S1 Checklist. STROBE statement—checklist of items that should be included in reports of observational studies.**
(DOCX)

**S1 File.**
(DOCX)

## Author Contributions

**Conceptualization:** Walton Wider, Jiaming Lin, Leilei Jiang.

**Data curation:** Walton Wider.

**Formal analysis:** Walton Wider.

**Funding acquisition:** Walton Wider.

**Investigation:** Walton Wider.

**Methodology:** Walton Wider.

**Project administration:** Walton Wider.

**Resources:** Walton Wider.

**Software:** Walton Wider.

**Supervision:** Walton Wider.

**Validation:** Walton Wider.

**Visualization:** Walton Wider.

**Writing – original draft:** Walton Wider.

**Writing – review & editing:** Jem Cloyd M. Tanucan, Jiaming Lin, Leilei Jiang, Lester Naces Udang.

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
