## [Decision Letter · Decision Letter 0]

23 Oct 2023

PONE-D-23-18382Who Makes a Better University Adjustment Wingman: Parents or Friends?PLOS ONE

Dear Dr. Wider,

Thank you for submitting your manuscript to PLOS ONE. After careful consideration, we feel that it has merit but does not fully meet PLOS ONE’s publication criteria as it currently stands. Therefore, we invite you to submit a revised version of the manuscript that addresses the points raised during the review process.

Kind regards,

Nkosiyazi Dube, Ph.D

Academic Editor

PLOS ONE

Journal Requirements:

2. You indicated that ethical approval was not necessary for your study. We understand that the framework for ethical oversight requirements for studies of this type may differ depending on the setting and we would appreciate some further clarification regarding your research. Could you please provide further details on why your study is exempt from the need for approval and confirmation from your institutional review board or research ethics committee (e.g., in the form of a letter or email correspondence) that ethics review was not necessary for this study? Please include a copy of the correspondence as an ""Other"" file.

Please also provide additional details regarding participant consent. In the ethics statement in the Methods and online submission information, please ensure that you have specified (1) whether consent was informed and (2) what type you obtained (for instance, written or verbal, and if verbal, how it was documented and witnessed). If your study included minors, state whether you obtained consent from parents or guardians. If the need for consent was waived by the ethics committee, please include this information.

Reviewers' comments:

Reviewer's Responses to Questions

**Comments to the Author**

1. Is the manuscript technically sound, and do the data support the conclusions?

Reviewer #1: Partly

Reviewer #2: Yes

2. Has the statistical analysis been performed appropriately and rigorously? 

Reviewer #1: No

Reviewer #2: Yes

3. Have the authors made all data underlying the findings in their manuscript fully available?

Reviewer #1: No

Reviewer #2: Yes

4. Is the manuscript presented in an intelligible fashion and written in standard English?

Reviewer #1: Yes

Reviewer #2: Yes

5. Review Comments to the Author

Reviewer #1: Thank you for the opportunity to review the manuscript. I found the manuscript insightful, however there are some areas that require elaboration, particularly the methods and results section. Below, I have provide specific feedback.

Introduction:

The section is well written. However, where the authors are making reference to studies carried out elsewhere, I recommend priority be given to recent studies. I came across few outdated sources, for example:

Line 43-45 and 69-71, since 1995 university culture has significantly transformed with many universities introducing various student support programmes to facilitate positive adjustment. The student population has also changed, for example - in many African countries we now have many students from previously disadvantaged groups entering universities. This on its own serve as a support system for the groups which were previously underrepresented in the universities.

Line 120-121, “most studies agree…” this statement need to be supported by more than 1 citation.

Methods:

The section can be better presented.

I struggled to identify information related to the study design, study setting, and participant sampling strategy.

Also, information related to the number of students invited to complete the survey, response rate, complete and incomplete responses is insufficient. I recommend the authors use the following subheadings to present the information:

1. Study design

2. Study setting

3. Sample and participant recruitment

Table 1: which faculty/school within the university where these participants drawn from? Literature has shown students experiences differs across faculties and schools.

Instruments:

“Section A consisted of questions about sociodemographic profiles, such as gender, age, 250 highest level of education, and ethnic group…” It is unclear what are you referring to? I suggest you have a section where you clearly outline the different sections of the survey, and present the participant demographic data as part of the results, so the reader can easily make sense of the survey sections and interpretation. ”

Pre-test:

What were the results or feedback from the students and academics?

Results and discussions:

The authors performed relevant tests, however the order is vague, particularly the interpretation. It is difficult is identify what where the key results for each of the studied variables.

The discussion is elaborative, however old sources are cited (e.g. Aries & Johnson, 1983; Collins and Repinski (1994); Fraley & Davis, 1997; Kenny, 1987). I recommend the authors attempt to cite recent articles.

I have noted data was collected in 2015, since then we have had COVID-19 which negatively affected universities and students' wellbeing across the world. Few studies have since been published, consider interrogating some of these studies, particularly those in line with your results.

Theoretical and practical contributions:

The contributions of the study are best covered as part of the discussion including policy implications.

Reviewer #2: Worth mentioning is that my research niches and the theoretical framework that I use are not clinical, quantitative research and psychology. I'm more into community development and related fields of practice. Nonetheless, I found this paper to have been excellently conceptualized, with logical themes that align the title, the aim, objectives, theoretical frameworks and methodology together. No doubt, these are mature and most likely academics/scholars. I have just a couple of suggestions: the authors must remember that the journal is international and so is their audience. As such the readers need context in respect some abbreviations and standards.

6. PLOS authors have the option to publish the peer review history of their article (what does this mean?). If published, this will include your full peer review and any attached files.

Reviewer #1: No

Reviewer #2: No

---

## [Author Response · Author response to Decision Letter 0]

28 Oct 2023

#Reviewer 1

1. Introduction:

The section is well written. However, where the authors are making reference to studies carried out elsewhere, I recommend priority be given to recent studies. I came across few outdated sources, for example:

Line 43-45 and 69-71, since 1995 university culture has significantly transformed with many universities introducing various student support programmes to facilitate positive adjustment. The student population has also changed, for example - in many African countries we now have many students from previously disadvantaged groups entering universities. This on its own serve as a support system for the groups which were previously underrepresented in the universities.

Line 120-121, “most studies agree…” this statement need to be supported by more than 1 citation. 

Answers:

- Thank you for the feedback. We updated the references in lines 43-45 and 69-71 to include recent studies and provided multiple citations for the statement on lines 120-121.

- Please refer to line 48, and 66-78.

- Please refer to line 127-128.

2. Method:

The section can be better presented. I struggled to identify information related to the study design, study setting, and participant sampling strategy. Also, information related to the number of students invited to complete the survey, response rate, complete and incomplete responses is insufficient. I recommend the authors use the following subheadings to present the information:

1. Study design

2. Study setting

3. Sample and participant recruitment

Table 1: which faculty/school within the university where these participants drawn from? Literature has shown students experiences differs across faculties and schools.

Answers:

- Thank you for your feedback. We've revised the method section to clearly include:

1. Study Design

2. Study Setting

3. Sample and Participant Recruitment

- Thank you for your query. In our study, we only collected data on the field of study and not the specific faculty or school within the university. We have included the information. Please refer to line 277.

- Details on the number of invited students, response rate, and complete/incomplete responses have also been added. Please refer to line 229-235.

Instruments:

“Section A consisted of questions about sociodemographic profiles, such as gender, age, 250 highest level of education, and ethnic group…” It is unclear what are you referring to? I suggest you have a section where you clearly outline the different sections of the survey, and present the participant demographic data as part of the results, so the reader can easily make sense of the survey sections and interpretation. ”

- Thank you for your feedback. We've revised the "Instruments" section to clearly outline the different sections of the survey. Please refer to line 258.

- In addition, we have moved the participant demographic data to the results for better clarity and understanding. Please refer to line 350-372.

Pre-test:

What were the results or feedback from the students and academics? 

Answer: We have included the feedback of students and academics. Please refer to line 325-328.

3. Results and discussions:

The authors performed relevant tests, however the order is vague, particularly the interpretation. It is difficult is identify what where the key results for each of the studied variables. The discussion is elaborative, however old sources are cited (e.g. Aries & Johnson, 1983; Collins and Repinski (1994); Fraley & Davis, 1997; Kenny, 1987). I recommend the authors attempt to cite recent articles. 

Answers:

- Thank you for the feedback. We've restructured the results section to sequentially outline the tests performed and to explicitly highlight the key findings for each studied variable. Please refer to 374-434.

- Thank you for the feedback. We reviewed the literature and incorporated more recent articles to ensure the discussion was current and relevant.

4. I have noted data was collected in 2015, since then we have had COVID-19 which negatively affected universities and students' wellbeing across the world. Few studies have since been published, consider interrogating some of these studies, particularly those in line with your results. 

Answers:

- Thank you for the feedback. We have revised the manuscript to incorporate recent literature, especially those addressing the impacts of COVID-19 on universities and students' wellbeing. Please refer to line 574-597.

5. Theoretical and practical contributions:

The contributions of the study are best covered as part of the discussion including policy implications. 

Answers:

- Thank you for the suggestion. We have revised the section to integrate the theoretical and practical contributions within the discussion. Please refer to line 598-632.

#Reviewer 2

1. Worth mentioning is that my research niches and the theoretical framework that I use are not clinical, quantitative research and psychology. I'm more into community development and related fields of practice. Nonetheless, I found this paper to have been excellently conceptualized, with logical themes that align the title, the aim, objectives, theoretical frameworks and methodology together. No doubt, these are mature and most likely academics/scholars. I have just a couple of suggestions: the authors must remember that the journal is international and so is their audience. As such the readers need context in respect some abbreviations and standards. 

Answer: Thank you for your valuable feedback and kind words regarding the paper's conceptualization and alignment. We appreciate the perspective you've provided from the community development field. Taking note of your suggestions, we have revised the manuscript to provide clearer context for abbreviations and standards to cater to our international audience.

---

## [Editor Report · Decision Letter 1]

7 Nov 2023

Who makes a better university adjustment wingman? Parents or friends?

PONE-D-23-18382R1

Dear Dr. Wider Walton,

We’re pleased to inform you that your manuscript has been judged scientifically suitable for publication and will be formally accepted for publication once it meets all outstanding technical requirements.

Kind regards,

Nkosiyazi Dube, Ph.D

Academic Editor

PLOS ONE

Additional Editor Comments (optional):

Thank you for the revisions and for paying attention to detail. This is an important study.
---

## [Editor Report · Acceptance letter]

11 Dec 2023

PONE-D-23-18382R1 

Who makes a better university adjustment wingman: parents or friends? 

Dear Dr. Wider:

I'm pleased to inform you that your manuscript has been deemed suitable for publication in PLOS ONE. Congratulations! Your manuscript is now with our production department. 

Kind regards, 

on behalf of

Dr. Nkosiyazi Dube 

Academic Editor

PLOS ONE